# ConnectUp: Co-Designing an Online Social Connection Platform with People with Disabilities and Carers

**DOI:** 10.3390/bs15101299

**Published:** 2025-09-23

**Authors:** Dominika Kwasnicka, Sophie Jokovich, Chelsea Atherton, Emily Joy, Genevieve Mullen, Joanne McVeigh, Stuart Jenkinson, Jessica Hatton, Paul Rogers, Ashul Shah, Enrique Mergelsberg, Craig Thompson

**Affiliations:** 1Centre for Health Equity, School of Population and Global Health, University of Melbourne, Melbourne 3010, Australia; 2Curtin School of Allied Health, Faculty of Health Sciences, Curtin University, Perth 6102, Australiajoanne.mcveigh@curtin.edu.au (J.M.); craig.thompson@curtin.edu.au (C.T.); 3Curtin enAble Institute, Curtin University, Perth 6102, Australia; 4Movement Physiology Laboratory, School of Physiology, University of Witwatersrand, Johannesburg 2193, South Africa; 5Carers Western Australia, Carers Association of WA, Perth 6000, Australia; stuart.jenkinson@carerswa.asn.au (S.J.); jessica.hatton@carerswa.asn.au (J.H.); paul.rogers@carerswa.asn.au (P.R.); 6Eduka Pty Ltd., Perth 6005, Australia; ashul.shah@eduka.com; 7Enberg Analytics Pty Ltd., Perth 6060, Australia; enrique@enberganalytics.com.au

**Keywords:** physical activity, social connection, loneliness, disability, digital health

## Abstract

People with disabilities (PwD) and carers report higher levels of isolation and lower physical activity (PA) participation than the general population. Yet, both PA and social connection are linked to improved health and quality of life. Innovative approaches are needed to address these disparities. This study aimed to (1) explore PwD and carers’ experiences with PA, social connection, and online platforms, and (2) examine their preferences and expectations for online tools supporting these needs. Six workshops (*N* = 17; 6 PwD, 6 carers, 5 both; aged 20–73 years, *M* = 52.94, *SD* = 15.21) involved the co-design of the *ConnectUp* platform prototype. Data were analyzed using reflective thematic analysis, identifying four key themes: (1) safety and vulnerability online; (2) disability inclusion and creating authentic social connections; (3) physical activity and disability; and (4) meaningful representation of PwD and carers on online platforms. Participants emphasized the need for platforms that foster genuine inclusion, real connections, and support for PA. The co-design process ensured these preferences shaped platform development. The *ConnectUp* prototype is currently being further co-developed and refined for national rollout, aiming to reduce isolation and promote wellbeing for PwD and carers through inclusive digital engagement.

## 1. Introduction

In Australia, one in six people live with a disability, which may be physical, intellectual, behavioral, or related to mental health ([5]). People with disabilities (PwD) experience poorer outcomes across multiple life domains, including employment, education, housing, and social support, embedding disadvantage into their daily lives ([5]). This systemic marginalization contributes to social exclusion and isolation, which in turn can limit participation in activities that promote wellbeing ([29]).

Carers face similar challenges. They often have fewer opportunities to engage in health-promoting activities compared to non-carers, due to caregiving responsibilities, emotional strain, and time limitations ([13]). The combined impact of living with a disability or providing care can significantly reduce opportunities for social connection and physical activity (PA) participation ([26]; [32]).

The benefits of regular PA are well established and include improvements in mental health ([39]), physical health ([36]), and cognitive functioning ([31]). Additionally, PA has been linked to increased social connection among PwD, contributing to enhanced overall wellbeing ([30]). Despite these benefits, participation in PA remains significantly lower among PwD and carers when compared to the general population ([4]).

A range of barriers contribute to this disparity, including environmental inaccessibility, negative attitudes from staff and other users at PA venues (e.g., gyms, recreation centers, sporting clubs), and a lack of knowledge about safe and suitable activities ([35]; [38]). Financial constraints and insufficient social support from family and friends further limit participation ([5]; [35]; [38]). Even when initial access is improved, maintaining long-term PA engagement can be challenging without ongoing motivation and support ([24]). However, individuals with strong social support and enabling environments are more likely to sustain behavior change ([24]).

Digital tools have increasingly been used to promote PA and build social networks, yet existing mainstream platforms (e.g., Facebook, Meetup, Strava) often fall short for PwD and carers. Research suggests that while these platforms may support connection, they frequently fail to provide safe, accessible, and inclusive environments tailored to the distinct needs of users with disabilities and carers ([2]; [12]). Digital platforms, including social media and mobile applications, are increasingly playing a role in enhancing social connection and promoting PA. Social platforms can help reduce geographical and environmental barriers by enabling remote communication and community participation ([42]; [27]). Apps like Bumble for Friends and other “people-nearby” applications allow users to form new connections within their local area, expanding their social networks ([18]; [33]; [37]). The COVID-19 pandemic further highlighted the potential of online tools, such as Zoom and Facebook, for helping PwD stay connected through structured virtual events ([40]).

Despite these opportunities, PwD often hesitate to use open-access platforms due to concerns about safety, previous negative experiences, and exposure to cyberbullying ([3]). This emphasizes the need for safer, more inclusive online environments. Platforms like *Alvie* and *Livewire* have been developed to address these issues for specific groups of PwD, yet there is currently no digital platform tailored to the needs of adult PwD and carers that supports both social connection and promotes PA engagement. To address these barriers, we developed a prototype of *ConnectUp*, a co-designed digital platform built specifically for adult PwD and carers. Unlike existing apps that target broad populations, *ConnectUp* was designed to reflect the lived experiences, communication styles, and accessibility requirements of its users.

Co-design was central to *ConnectUp*’s development, enabling meaningful participation from PwD and carers throughout the design process. Prior research has demonstrated that involving PwD and carers directly in the design process enables the development of platforms that reflect their specific needs, values, and lived experiences, promoting genuine inclusion from the outset ([43]; [25]). Although co-design can be time consuming, it is often cost-effective in the long term by reducing the need for redesign ([16]; [23]). An online platform designed specifically for PwD and carers has the potential to provide a safe, inclusive, and engaging digital space ([41]).

In response to this need, *ConnectUp* was developed in collaboration with Carers WA. The online platform aims to foster social connection and promote participation in PA among PwD and carers. To inform the design of the *ConnectUp* prototype, this study aimed to: (1) explore PwD and carers’ experiences with PA, social connection, and online platforms, and (2) examine their preferences and expectations for online platform supporting these needs. These insights directly informed the design of the *ConnectUp* platform prototype.

## 2. Materials and Methods

### 2.1. Research Design

This study employed a phenomenological approach to explore the lived experiences of PwD and carers in online environments ([34]). These past experiences shaped participants’ future consciousness, their ability to anticipate, imagine, and form expectations about future possibilities which informed their preferences and expectations for the prototype *ConnectUp* platform ([1]). Co-design workshops were conducted to explore PwD and carers’ experiences with PA, social connection, and online platforms and to collaboratively develop the platform, ensuring that the design process was grounded in the real-life experiences, needs, and insights of PwD and carers.

### 2.2. Participants

Six workshops were conducted with 17 participants (male = 8, female = 9; PwD = 6, carer = 6, and PwD and carer = 5) aged 20 to 73 years (*M* = 52.94, *SD* = 15.21). Participants’ disabilities included physical, intellectual, and mental health conditions. Participants who were carers had different caring responsibilities ranging from ad hoc caring to full time caring roles. Workshops included a mix of carers, PwD and people identifying as both. We conducted six workshops (Table 1).

Participants were included in the study if they were adults (aged ≥18) and identified as a PwD, a current carer, or both and had to understand and speak English. Participant recruitment was facilitated by Carers WA (a non-profit, community-based organization and registered charity dedicated to improving the lives of carers living in Western Australia) and utilized a convenience sampling approach, which included carers representatives’ network, Carers WA social media, and word of mouth. Carers WA was the lead organization responsible for delivering this project and developing the *ConnectUp* platform.

The study was approved by the Institutional Review Board by the Human Research Ethics Committee at Curtin University (HRE2021-0678), Perth, Australia. All participants provided informed consent prior to participation.

### 2.3. Materials

Participants completed pen-and-paper demographic questionnaires, including questions on age, gender, whether they are a carer or PwD, and type of disability if they identified as having one.

Workshops 1–4 used a predetermined set of questions (discussion topics, Appendix A) to target specific content and engage participants. Workshops 5–6 used a draft prototype of the *ConnectUp* platform (Figure 1), to facilitate direct feedback and recommendations to platform developers. The prototype draft was showed as clickable mobile phone application generated in Figma (v. 2020). The workshop facilitator showed the user’s journey screen-by-screen, asking workshop participants to share their feedback and reflections on the demonstrated prototype, including look-and-feel, wording, and functionality. Workshops averaged 92 min in duration (*M* = 92.34, *SD* = 15.28).

### 2.4. Procedure

PwD, carers, and key stakeholders (workshop 6) were invited to attend workshops conducted by Carers WA. Workshops 1–5 were facilitated by a person with a disability (SJe) with the support of a behavioral researcher (DK). A web design specialist (AS) facilitated workshop five; workshop six was facilitated by a behavioral researcher (DK). Participants were asked about their experiences, beliefs, and attitudes towards improving social connection, PA, and general wellbeing. They were also asked to elaborate on online platforms that they used and to share their preferences and expectations for online tool supporting these needs. The workshops were very informal and participants were encouraged to share their experiences and to reflect on how online platform could support their needs to improve social connection, inclusion, and encourage them to engage in shared PA with other PwD and carers. The initial two workshops were held at Cityplace Community Centre in Perth, WA. Data were digitally audio and video recorded. Workshops 3–6 were held online via Zoom, which enabled digital audio-visual recording. Workshop recordings were then transcribed using Otter.ai, an online transcription tool.

### 2.5. Analysis

Reflexive thematic analysis ([8], [9]) was completed using NVivo software and applying conservative methods such as ‘sticky notes’ and hand-drawn mind maps. The transcripts were analyzed in six-stages: (1) familiarizing oneself with data; (2) generating initial codes; (3) searching for themes; (4) reviewing themes; (5) defining and naming themes, and (6) producing the report. Stages 1–3 involved four researchers (SJo, CA, EJ, GM). Stage one involved reviewing audio-visual and transcript data to establish familiarity. Within stage two, the researchers independently coded the transcript data using NVivo software. Stage three involved the initial search for themes discussing their independent coding to establish and achieve consensus on comparable concepts as themes developed. During stage four and five, the four researchers (SJo, CA, EJ and GM) who coded the data engaged in peer review of the initial themes and defining and naming themes with three experienced researchers (JMV, CT and DK). All researchers participated in stage six, the write-up and development of the thematic narrative. To ensure the trustworthiness of the coding and theme development process, researchers engaged in regular team meetings and an audit trail for code and theme development was maintained. To ensure that the prototype of the *ConnectUp* platform could meet the needs of PwD and carers, the research team provided participants with the opportunity to provide feedback on the findings.

### 2.6. Researcher Characteristics and Reflexivity Statement

During this research project four researchers (SJo, CA, EJ and GM) were Occupational Therapy Honors students and they led data analysis and reporting under the supervision of JMV, EM, CT and DK. JMV is a physiology researcher with expertise in measurement and analysis of physical activity in various populations. SJe identifies as living with a disability, working for Carers WA in the system policy and strategy role. As a person with a disability, SJe led the facilitation of the co-design workshops. During this research project, JH was working for Carers WA as a community capacity development officer, together with PR who is a lead team manager at the same organization. SJe, JH and PR, led consumer recruitment for the project. AS is a computer engineer and web design specialist with experience developing IT solutions for non-government organizations, and he led the development of the online platform; he also identifies as a PwD. The involvement of researchers with lived experience of disability (SJe, AS) and staff from Carers WA (SJe, JH, PR) informed the design of the co-design workshops to ensure cultural and contextual relevance. This also supported more meaningful participant engagement and trust-building during recruitment and facilitation.

EM is a health psychologist who has experience working with individuals with various lived experiences, including living with diabetes and mental health challenges. CT is an occupational therapist with research and clinical experience working with people with physical and mental health conditions. DK is a behavioral scientist with expertise in digital health. DK co-facilitated the co-design workshops with SJe. The mix of disciplinary expertise, occupational therapy, behavioral science, health psychology, and physiology, shaped the development of the interview guide, coding framework, and interpretation of the data. For example, CT brought clinical insight into how disability and caregiving influence daily life, while EM and DK contributed knowledge of behavior change strategies. This diversity enriched the analysis but also required ongoing reflection during team discussions to address disciplinary biases and avoid privileging one perspective.

## 3. Results

The study results informed the design of the *ConnectUp* platform. Four themes were identified in the data (Figure 2 shows the intersection of themes and subthemes). They were: (1) safety and vulnerability online; (2) disability inclusion and creating authentic social connections; (3) physical activity and disability; and (4) meaningful representation of PwD and carers on online platforms.

### 3.1. Theme 1 Safety and Vulnerability in the Online Environment

Safety was the most prominent topic within each workshop and was emphasized by all groups (PwD and carers). Safety was identified as the foundation for PwD to authentically engage in an online space. The safety theme incorporated privacy, cybersecurity, social safety, physical safety, psychological safety, and financial protection. Participants articulated a broader sense of vulnerability within the online community. Three sub-themes were identified within this theme.

#### 3.1.1. Potential Risks and Issues Present When Someone with a Disability Is Online

Reports of abuse, harassment, or coercion were commonly reported experiences of PwD when engaging in online social connections. Having a cognitive disability was presented by participants as potentially having an increased risk of negative experiences such as online scamming. Additionally, many participants discussed online bullying and discriminatory comments that caused distress and made them feel unsafe online. These issues were noted despite many participants utilizing online systems to improve safety; one participant reflected:
*One of the issues was lots of scammers calling him and saying to do this, and I keep saying to him: “nobody’s gonna call you like that”. But his level of ability to make decisions is not there anymore.*Participant 13 (Carer)

#### 3.1.2. There Is No ‘One Size Fits All’ for Safety

Issues relating to safety varied depending on the type or impact of disability. Concerns from participants with mental health conditions highlighted the potential for emotional triggers when socializing with others and the difficulty of communicating that, especially online.

Participants discussed the need to balance autonomy and support from carers when communicating online, especially for those with intellectual disabilities. Furthermore, it was noted that using online platforms can be difficult due to social behaviors and the fear of miscommunication.
*I think it’s hard. Particularly if you’ve got some sort of intellectual disability. With those chatlines, you have the potential to say things you don’t really mean to say, or they’re misinterpreted or some other thing, and there can be a real danger there.*Participant 2 (PwD)

#### 3.1.3. Safety Mechanisms Are Necessary for Online Platforms to Lower Risks

Discussions about safety and mitigating risk for PwD utilizing online platforms were recurrent throughout all workshops. To mitigate risk and support safety education and awareness, participants suggested comprehensive, accessible guides covering topics such as privacy, online safety, and considerations for meeting people in person.

User-friendly language accessible for all disabilities, LGBTQI+ populations, and culturally diverse groups was identified as a basic consideration to support inclusion. When abuse or safety concerns become apparent, participants suggested that the platform moderators should be reactive, enabling reporting of harassment or hurtful language. Finally, carers expressed that guardian access features could be beneficial in supporting those with disadvantaged insight, judgment, or communication abilities.
*My daughter is not on Facebook, she’s not on Instagram, she doesn’t have the capacity to be on those... I’d love to be able to create the profile of who she is as an individual. But it be that I’ve got ownership over it, because she’s not going to be the one putting in the preferences, it’s going to be me, just talking on her behalf. I’m advocating for her, but I think it’s important that she, as an individual, has her identity.*Participant 2 (Carer)

### 3.2. Theme 2 Disability Inclusion and Creating Authentic Social Connections

This theme reflected PwD and carers’ desire for social connection and inclusion. The discussions focused on previous social experiences, including positive and negative experiences in social situations. Participants also stated that inclusivity influenced whether a social experience would be positive or negative. Many participants agreed that an inclusive environment facilitated opportunities for social connection. However, labeling social groups as ‘inclusive’ did not automatically mean social connection would occur.
*Sports organizations across the board now often say they’re inclusive, they all have inclusive policies… But my daughter for example, does mainstream netball and does ‘No Limits’ netball, which is a disability contest. ‘No Limits’ netball is her ‘fun netball’. She gets there, she hugs the girls, they run up to her. She does mainstream netball; it’s more competitive netball and those girls don’t talk to her.*Participant 2 (Carer)

Participants expressed challenges in establishing social connections for PwD noting that developing a platform such as *ConnectUp* will allow PwD and carers to connect with each other and engage in PA together, further building on their peer network. The participants suggested that the platform should be inclusive by offering personalized content tailored to users’ interests, goals, and lived experiences, for example, by recommending relevant activities or groups.
*These* [individuals] *are going to grow older, they need their own friend groups, they can’t just rely on us or the carers all the time. They need people that want to do like minded things. They need those connections to keep them going through life as they get older.*Participant 2 (Carer)

PwD reported anxiety about possible misinterpretations of social cues, which can lead to miscommunications on online messaging platforms. This generated further recommendations for the *ConnectUp* platform, including communication prompts to assist participants with appropriate questions and suggestions while using the chat function.
*I think having support in helping with language because sometimes I struggled in school, and I struggled with language. I think having a* [tool] *that you can click on that says ‘this is what you could suggest’ or ways of helping with the communication. Like prompts and stuff like that because I think there’s so much pressure—I* [often] *don’t know what to say.*Participant 13 (PwD and Carer)

Awareness of barriers experienced by PwD was raised as an important aspect of creating authentic social connections and improving disability inclusion. Participants discussion focused on PwD’s barriers in accessing and participating in social connections and PA. These barriers included the lack of awareness in terms of venues, public places, and society in creating an accessible and inclusive environment for PwD.
[People can be] *scared about hurting people with disability, so they just don’t want to play with them. Because they’re worried that ‘oh if we touch her, she’ll break something’. Or ‘if we push her too hard, she’ll fall, and we’ll get in trouble for that’. So, they just kind of segregate you.*Participant 13 (PwD and Carer)

### 3.3. Theme 3 Participation in Physical Activity for PwD and Carers

This theme refers to the PA topics identified in the discussions and has two subthemes: (a) differing perspectives of PA and disability and (b) barriers to PA participation.

#### 3.3.1. Differing Perspectives of Physical Activity and Disability

Participants were aware of the benefits of PA and generally held positive intentions towards participation. However, experiences of PA participation varied between individuals. Some had positive experiences within both disability-exclusive and non-exclusive PA types; however, experiences were more often negative.
*She is very reluctant to participate in sports. I think that’s more of the growing up process of being rejected. I think it’s quite deep within her now that sport often means ridicule, and so it’s hard to get her involved.*Participant 4 (Carer)

Participants with physical disabilities or those who expressed difficulties with social judgment often had negative experiences from injuries from PA. Participants expressed issues with a lack of insight into physical limitations and safety from others during contact sports.
*We want to compete, but other people don’t want to challenge you and even when you can do more... but then you’re also more vulnerable physically.* Participant 14 (PwD)

Additionally, some participants noted that past experiences of bullying and exclusion while participating in social sports had resulted in negative associations with PA. For participants with mental health conditions, these past experiences heightened anxieties towards participation in PA, resulting in further avoidance.
*Growing up, sports has always been this very segregating experience because I* [thought]*, oh, you can’t do sports. Well, you sit outside.*Participant 15 (PwD)

#### 3.3.2. Environmental, Social and Financial Barriers to PA

Participants reported environmental barriers more frequently than social barriers to PA participation. A prominent environmental barrier was that participants did not know what PA options were available within their community. Consequently, participants reported that they “missed out” on activities that have occurred due to a lack of knowledge about activities, places, and events and limited available resources with regularly updated information.

Participants also noted that the National Disability Insurance Scheme (NDIS) could be perceived as a barrier, and they expressed confusion in claiming social or recreational supports to engage in PA. Participants reported that they were concerned about needing support in the community and their confusion about the usability of companion cards (for PwD who require attendant care support to participate in community venues).

Furthermore, people in receipt of disability support payments noted additional concerns about the costs associated with PA, which posed as a further financial barrier. Participants also reported that they perceive the lack of time to participate in PA due to their frequent appointments, changes in energy or health, and restricted funded hours of support work. Consequently, participants felt they either had no time for recreation or, in contrast, too much time spent being sedentary.
*I feel as a carer, the person I care for some days, there are appointments from the time you get up... and so there is no time for anything really. But then other days, there’s nothing to do. So, you’ve got to grab your opportunity when you can. So, I think it’s different for everybody.*Participant 1 (PwD and Carer)

When people were able to find suitable activities that were financially and logistically feasible, social barriers impacted PA participation. Participants agreed that they felt more motivated to participate in PA when they had someone else with them, but many did not have companions in these contexts. Participants reported that this often led to decreased motivation and reduced PA participation. In contrast, prospects of socialization within PA participation were met with enthusiasm and excitement from participants.
*You might have a group of 10 people that meet up, that you have already talked to you beforehand, and you go there feeling more confident... Knowing I’m not the only person with inaccessible needs going. And because we’re all going together. There’s that feeling of solidarity.*Participant 16 (PwD and Carer)

### 3.4. Theme 4 Including PwD and Carers on an Online Platform

This theme outlines what PwD wanted to include in the online platform and contains two subthemes, (a) platform design features and (b) user experience perspectives.

#### 3.4.1. Accessible Online Platform Design Features

Accessibility was described as the key foundation to the design of the platform. Participants recognized that the accessibility needs differ amongst PwD and suggested the options of voice-to-text and hands-free to be incorporated features within the platform.
*His hands get cramps, or they don’t always work to type even though we have got a special mouse… So that will be something useful if they can use it with the words to respond to the actual speech.*Participant 13 (PwD and Carer)

Participants noted that some PwD use assistive technology for online access, but not all online platforms are compatible for their use. Participants indicated that the platform should be adaptable and compatible with a wide range of devices, including those using assistive technology.

“Keep it simple stupid” was a common phrase used to describe the design of the platform itself, championing the need for a simple online interface. To enhance accessibility, participants suggested that the platform should be modifiable to suit the users’ preferences, such as options to zoom, greyscale, increase font sizes and picture imageries for the visual presentation of the platform, and voice control features for navigation.

#### 3.4.2. User Autonomy in the Online Experience

Participants indicated that to foster social connection through the platform, autonomy would need to be the key component in platform design. Participants need choice in dictating how they navigate the platform and develop new connections with others. To do so, options to set preferences around ages, genders, interests, and travel distances are important features to include in the design.
*I feel safer selecting female. So, to be able to have that option of saying hey, I’d just feel more comfortable meeting with another girl.*Participant 16 (PwD and Carer)

When shown the set-up of a personal profile in the prototype, participants stated that they did not like “dropdown boxes and forms”. Participants argued that such methods can trigger anxiety, because PwD regularly complete forms, e.g., service agreements. Participants instead suggested that the profile should incorporate a series of easy to answer, quizzes, short questions and some open text fields.
*I’ve been having to fill out a lot of online forms lately, so got some experience. And so many of them drive me nuts because they have, you know, anything with prefilled boxes, we don’t all fit into boxes. I come across them all the time. And it’s like, you cannot find the answer that fits you.*Participant 16 (PwD and Carer)

Some participants said it would be beneficial to allow users the option of taking a break from the platform. Participants recommended a simple “this person is taking time away from the platform for now” message within chats or individual profiles.
*I [have] anxiety. I might feel social one minute, and then tomorrow I might not feel social. Could there be a pause button? So, I can take myself off and on.*Participant 15 (PwD)

Participants noted that the technology literacy of PwD could be a limiting factor to using the platform. Many participants indicated that their technology literacy was low. Lower technology literacy was thought to have a negative impact on participants’ perspectives on platform development. Participants stated that they were hesitant to provide basic information required to join the platforms, such as emails and personal information, which is often necessary for authentication within any platform. Overall, study participants were enthusiastic about the platform’s development, which will allow them to connect with people like them and engage in different forms of PA at the convenient time and in the most suitable location. They also expressed their enthusiasm that they are involved in the platform development early on and that their voices are “being heard” when the features and functionalities are being designed.

## 4. Discussion

This study adds to the growing body of literature on accessible digital design by centering the perspectives of both PwD and carers in the co-design of a platform aimed at promoting PA and social connection. While prior research has explored barriers to PA participation and use of digital platforms separately, few studies have addressed the intersection of these issues through a co-design approach that includes both groups. Our findings offer novel insights into user preferences, trust-building features, and perceived affordances that are often overlooked in mainstream platform development.

This study was conducted to explore PwD and carers’ experiences with PA, social connection, and online platforms and to develop a prototype online platform to foster social connection and PA. The results contribute both to the design of *ConnectUp* and to wider conversations about digital inclusion and social connection. Four interconnected themes were identified in the analysis: (1) safety and vulnerability in the online environment, (2) disability inclusion and creating authentic social connections, (3) participation in physical activity for PwD and carers, and (4) including PwD and carers on an online platform. that the provision of adequate safety mechanisms must be at the forefront of online platform development to enable authentic and safe engagement. The identified barriers to PwD engaging in PA included social exclusion, potential harm, environmental issues, and lack of knowledge of available activities. Participants expressed the need for a simple, easy-to-use, non-confrontational platform that enabled universal accessibility. Finally, PwD and carers indicated their desire for social connection and genuine inclusion would motivate engagement in the *ConnectUp* platform, which may assist in creating authentic social connections and participation in PA.

Social exclusion for PwD can be attributed to misconstrued societal judgements and/or ableist perceptions ([28]). Aligned with previous research the findings from this study emphasized the exclusion of PwD within mainstream sports, whilst experiences with disability-inclusive sports were perceived more positively ([10]). This can be attributed to building further connections within an environment of people with similar backgrounds, interests, or needs ([19]). Social inclusion and connection are associated with social opportunities and can simultaneously enable new social connections for PwD ([14]). An importance of developing an inclusive online environment to enable social connections in PwD and carers was highlighted in this study. PwD may encounter barriers to inclusion when such needs are not considered in the design phase ([7]).

The reality for many PwD are challenges arising from living in a society that is designed for people without disabilities, and the inclusion of PwD may be often an afterthought ([14]). The perspectives from the participants of this study suggested that PwD commonly felt a lack of inclusion due to the limited accessible environments, social exclusion, difficulty establishing and maintaining social connections, and insufficient knowledge about disability within the general public ([6]). These environmental barriers are commonly associated with negatively affecting participation in PA for PwD ([35]; [38]). The findings of this study and previous research highlighted the difficulties in participating in PA due to energy levels, lack of time, or financial restraints ([20]).

Social media can be used to transcend such environmental barriers by providing instant access to broader social connections ([11]). Consequently, social media has the potential to allow PwD to navigate non-inclusive physical environments, which often involves intricate logistical planning. While online social platforms can reduce some barriers to inclusion, the vulnerability of PwDs regarding online safety is another barrier that needs to be considered ([17]). Previous harassment and emotional harm experienced online by PwD and carers often leads to them being hesitant to use online platforms ([2]). Fostering accessibility should be at the forefront of designing the online platform to ensure safety and usability for the target user group of PwD ([15]). Technology has reduced many barriers for PwD, but special consideration is required to prevent the creation of new challenges in the process of developing online platforms.

The design concept of universal design benefits the general population and is not limited to PwD, it allows flexibility and autonomy in how users interact online ([23]). This incorporates choice when creating profiles and a filtering system within the user matching component, with safety components and accessible design at the forefront ([25]). These findings are consistent with previous literature identifying accessibility features, including enhancing graphic design to simplify the user experience ([21]). To achieve accessibility when designing an online platform, PwD and carers must be included and have a voice during the initial and implementation stages of design. A co-design process gave voice to PwD and carers to stipulate their wants and needs, facilitating true inclusion. Incorporating participants in the co-design process from the early stages was necessary in achieving true inclusion. Whilst it may result in a longer process initially, including the targeted user group in co-design can be more efficient and cost-effective as it enables a universal design from the outset ([16]; [23]).

A key design decision in the development of the platform was to limit access to persons with disabilities (PwD) and carers. This choice emerged directly from user consultations, where participants expressed a strong preference for a dedicated space tailored to their specific needs and lived experience. While concerns have been raised about potential exclusion, participants emphasized that existing mainstream platforms often fail to provide environments where they feel fully understood or adequately supported. Aligned with other researchers, we acknowledge the complexity of dedicated versus mainstream platforms and the potential for unintended consequences of separate spaces ([22]). However, the creation of a closed, community-specific platform was intended to promote psychological safety, foster peer-to-peer support, and facilitate communication without the burden of explaining disability-related experiences or carer-experiences. Future iterations may consider broader integration; however, the initial focus remains on addressing the unique priorities identified by the core user group.

This study has several strengths. First, it allowed PwD and carers to participate in designing the prototype online platform to meet their needs and wants. Second, through workshop discussions and co-design, this study offers insights into PwD’s and carers’ lived experiences of inclusion and accessibility. Third, the findings highlight the importance of having the appropriate safety mechanisms, accessible features and autonomy as key features to enable an inclusive online platform for PwD and carers.

The findings of this study need to be considered in line with the following limitations. Although the aim was to recruit a diverse sample of PwD and carers in terms of disability type, the included sample was not as diverse as anticipated (e.g., people with limited hearing or limited vision did not participate) likely due to the recruitment approach which may have influenced the range of perspectives captured. This study did not explore intra-disability diversity in depth, and future research is needed to examine how differing lived experiences may shape preferences and needs. During workshops, there was a potential for social desirability bias, and participants potentially provided socially desirable opinions and insights; however, the results did reflect a diversity of opinions. Finally, member checking was inconclusive due to the low response rate (only one person participated).

Further research is needed to understand the relationship between purposefully designed online platforms for PwD and carers and its influence on social connection through PA. Also, there is a need to pilot, implement, and scale the *ConnectUp* to fully understand this tool’s effectiveness, cost-effectiveness, and maintenance. The proposed online platform can potentially improve participation in PA and social connections for PwD and carers. The platform is currently further co-designed and it will be rolled out nationally in Australia in 2026.

## 5. Conclusions

Participants’ insights offered critical guidance on the features that contribute to a user-friendly online platform for social connection among PwD and carers. A strong emphasis was placed on safety, reflecting the vulnerability many PwD experience in online environments. Equally important was the need for genuine inclusion, both in platform design and user experience which was seen as essential for fostering motivation and enabling authentic social connection. These priorities speak to wider concerns in digital design, where accessibility and inclusion are increasingly recognized not only as technical requirements but as fundamental to promoting digital equity and social participation. We recognize, however, that broader societal attitudes and systemic barriers remain challenges that technical accessibility alone cannot fully resolve.

Participants also identified environmental barriers as the primary obstacles to engaging in PA. Common challenges included a lack of accessible venues, limited information about available activities, and experiences of social exclusion. These barriers often compounded the difficulty of maintaining regular PA. To address these issues, participants recommended several features to enhance accessibility and usability. These included autonomous control over user profiles, the ability to pause accounts, integrated communication prompts, a customizable interface, and voice control for navigation. Incorporating such elements into platform design can significantly improve user experience and ensure the platform meets the diverse needs of PwD and carers. More broadly, these findings highlight how digital platforms can play a role in shaping wellbeing, not only by facilitating access to PA but also by enabling meaningful social interaction. By aligning design features with the lived experiences of PwD and carers, mainstream platforms and apps can move toward more inclusive digital environments that foster both autonomy and belonging.

## Figures and Tables

**Figure 1 behavsci-15-01299-f001:**
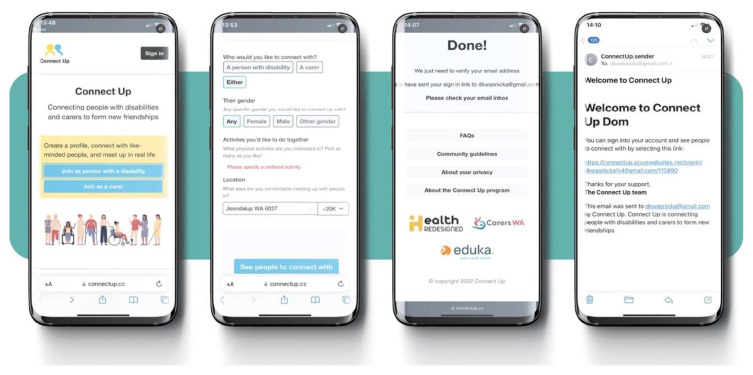
Screenshots of prototype of the *ConnectUp* platform that was developed based on the workshop insights.

**Figure 2 behavsci-15-01299-f002:**
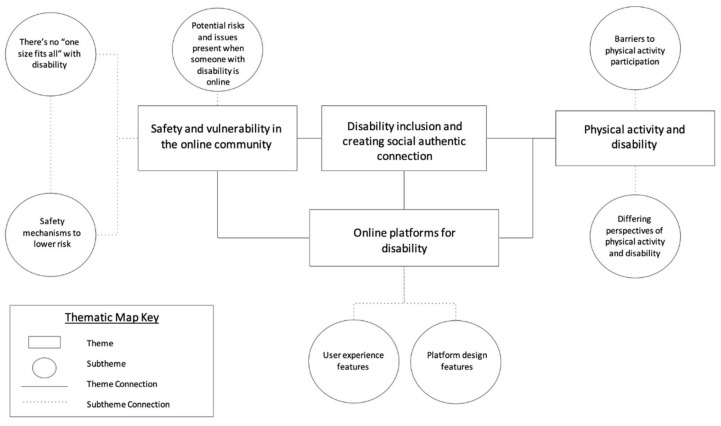
Key themes and sub-themes derived from the workshops.

**Table 1 behavsci-15-01299-t001:** Number of workshops, participants and objectives of each workshop.

Workshop Number	Participants	Objectives
1–4	PwD *n* = 6 Carer *n* = 6 Carer and PwD *n* = 5 (Total *N* = 17)	To explore participants’ experiences, perspectives and opinions of social connectedness, engagement in PA, and the use of online platforms. To collect information regarding the participants’ preferences and expectations of an online platform, and the perceived barriers and enablers to physical activity participation and social interaction.
5	PwD *n* = 2 Carer *n* = 3 (Total *N* = 5)	To explore participants’ feedback and perspectives on the prototype online platform (developed based on workshops 1–4).
6	Representatives of the community organizations (Total *N* = 13)	To gain interest and feedback regarding the preliminary prototypes from potential adopters of the platform. To discuss potential ways to promote, implement and scale the platform. (The results of workshop 6 are reported elsewhere)

Note: Some of the participants from workshop 5 also attended workshops 1–4. This article reports results of workshops 1–5 only. Workshop 6 informed further successful funding applications for national roll-out of the *ConnectUp* platform.

## Data Availability

Data is available on request from the study authors.

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
