# Peer review of "ConnectUp: Co-Designing an Online Social Connection Platform with People with Disabilities and Carers"

_behavsci, 2025, doi:10.3390/bs15101299_

Round 1

Reviewer 1 Report

Comments and Suggestions for Authors

Dear Authors,

Thank you for your work in conducting this research and preparing this article. The topic exploring PwDs, carers, physical wellbeing, health, and the role of digital media is both timely and important. There is significant potential here to advance scholarly discussions and shape future research that can drive meaningful change.

However, in its current form, the article has several limitations that, if addressed through substantial revision, could elevate it to a stronger contribution to the field which you clearly have the capacity to achieve.

Areas for Improvement

Introduction
The paper positions ConnectUp as its starting point, but provides minimal context about the project and prototype. While it may be familiar to audiences in Australia or readers of your other publications, readers of this article need more information to fully grasp its uniqueness. In a revision, I recommend clarifying what ConnectUp is, what distinguishes it from existing platforms, and why PwDs and carers require a separate digital space for exploring PA options. Is there existing research on how PwDs and carers use mainstream platforms (e.g., Facebook, local meetup apps) for these purposes? What barriers do they experience which is unique to this group which ConnectUp will address?

You note that “there is currently no digital platform tailored to the needs of adult PwDs and carers that supports both social connection and PA engagement,” but it remains unclear what specific needs are unmet by mainstream platforms. Are these related to negative online experiences, safety concerns, or cyberbullying? Many of these issues are not unique to PwDs, so clarifying what makes this population’s needs distinct would strengthen your argument.

Your point about co-design is convincing and draws on the important work of previous scholars such as Elizabeth Ellcessor’s work (Restricted Access), you may want to consider reading this which examines how form, experience, and the role of designers shape digital spaces for PwDs and when one aspect is missing how it creates unique barriers for PwDs.

Methods
In Section 2.3, you note that workshops 5–6 used a draft ConnectUp prototype, but provide little detail about what the prototype entailed. What did participants do with it? What features were tested? How long did they interact with it? Did they receive guidance or explore it freely? This goes back to the original point of providing context to ConnectUp for the following arguments to make sense to the reader.

Much of the methods section focuses on which author facilitated which activity (writing in this way also makes the narrative read in a clunky way), but this does not help readers understand the process from the participants’ perspective. More detail is needed on how participants engaged in the workshops and what their involvement was like.

You mention that Carers WA was the recruitment site but do not explain why this setting was chosen. Given that a member of the research team (PR) is also a manager at Carers WA, how did this influence recruitment and participants’ willingness to participate?

The description of the research team’s identities and skills (e.g., DK as a behavioural scientist, CT as an occupational therapist) is valuable but currently disconnected from a discussion of reflexivity. Clarify how the team’s backgrounds shaped the study design, participant interaction, rapport, recruitment, and potential biases during facilitation and your interpretation of the findings. This is important as only one member engaged in member check.

There is also no mention of research ethics and a note on this would help the reader to understand how you ensured ethical research took place and where ethical approval was obtained.

Results
Although the study’s focus is on co-designing ConnectUp, the results only briefly address this. Most findings describe broader themes such as online safety and inclusion. It may be worth reconsidering whether the paper should be framed more broadly around the features and social affordances that PwDs and carers want from digital platforms to support PA and social connectedness. You might find relevant perspectives in Kaur & Saukko’s work on how PwDs use mainstream platforms instead of those explicitly designed for them, sometimes dedicated spaces can unintentionally reinforce exclusion as many PwDs are found helping each other overcome digital design barriers in support groups for people with disabilities.

There is also occasional confusion due to the combined analysis of two population groups: PwDs and carers, without clear distinctions between them. At times when mentioning ‘participants’ it is unclear which group is being referenced. Moreover, the diversity within the PwD community and within one community (intra-disability diversity) is not acknowledged; P. Tsatsou’s work may be helpful here. Treating PwDs as a single, homogenous group while only acknowledging the conditions risks oversimplification and shows a lack of awareness of literature in disability studies.

It would be helpful to clarify whether individual workshops included both PwDs and their carers together. Some quotes (section 3.1.3) suggest that carers were often parents who are speaking about young children with disabilities, but there is little critical analysis of how carers’ perceptions may frame disability in infantilising or limiting ways, despite PwDs’ own sense of agency and capability. This tension is well documented in disability studies and should be critically addressed.

Some themes, such as “fear of miscommunication” (section 3.1.2), appear unrelated to the unique needs of PwDs and carers and could apply to digital platform users more broadly. Similarly, section 3.2 notes that the platform “could be inclusive and guided by the person’s interests and experiences,” but without clear context about the platform’s features, this is vague and makes little sense to the reader.

Discussion
It is not clear what the article’s original contribution is or what new insight it offers that advances debates about accessible digital platform design for promoting PA among PwDs and carers. Please clarify the main argument and how it builds on, contrasts with, or extends existing scholarly literature. I would also suggest updating the abstract in line with this.

You note that the results did not reflect a diversity of views, but do not explore why. I have the impression this was due to the sampling strategy, or was this due to other factors which are not discussed? Discussing this would help situate the findings and acknowledge potential limitations with clear explanations in a convincing way.

Conclusion
In the conclusion, you write about the need for “genuine inclusion … essential for fostering motivation and enabling authentic social connection,” but it is unclear who these connections are between: PwDs and other PwDs, carers and PwDs, or wider community members? Prior work (see, Kaur & Saukko) suggests that even when digital platforms are technically accessible, social connections are still hindered by broader societal attitudes towards disability when the connection is between PwDs as well as between PwDs and able-bodied people. Will the ConnectUp platform address such existing barriers presented in current research?

Finally, the paper implies that PA is an activity to be sought externally, yet many PwDs and carers may integrate PA into daily routines at home, at work, or in social contexts. How did participants conceptualise PA and what does this look like for PwDs and carers? Although the interview guide included relevant questions about this, I anticipate the workshops probably answered this, but the article does not clearly present these insights. Therefore overall the connection between PwDs, carers and PA is loose.

Overall, the study’s findings about what PwDs and carers want in a digital platform appear transferable to mainstream platforms and apps. A revised conclusion could situate these insights within broader discussions about accessibility, digital design, social relations, and everyday wellbeing, which could help clarify the article’s contribution and relevance.

Overall, I appreciate it is not easy to bring many elements together in a single article, and it is therefore down to the authors to rethink how to reframe the main argument so the contribution is clearer. Whether it is making ConnectUp the focus or broader issues around digital accessibility and co-design. I wish the team the best with figuring this out and hope the comments here support you with the revision.

Author Response

Comment 1: Dear Authors,

Thank you for your work in conducting this research and preparing this article. The topic exploring PwDs, carers, physical wellbeing, health, and the role of digital media is both timely and important. There is significant potential here to advance scholarly discussions and shape future research that can drive meaningful change.

However, in its current form, the article has several limitations that, if addressed through substantial revision, could elevate it to a stronger contribution to the field which you clearly have the capacity to achieve.

Response 1: We would like to thank the reviewer for their positive feedback and for providing constructive feedback to enable us to improve the manuscript to meaningfully contribute to the field.  

Areas for Improvement

Comment 2: Introduction
The paper positions ConnectUp as its starting point, but provides minimal context about the project and prototype. While it may be familiar to audiences in Australia or readers of your other publications, readers of this article need more information to fully grasp its uniqueness. In a revision, I recommend clarifying what ConnectUp is, what distinguishes it from existing platforms, and why PwDs and carers require a separate digital space for exploring PA options. Is there existing research on how PwDs and carers use mainstream platforms (e.g., Facebook, local meetup apps) for these purposes? What barriers do they experience which is unique to this group which ConnectUp will address?

You note that “there is currently no digital platform tailored to the needs of adult PwDs and carers that supports both social connection and PA engagement,” but it remains unclear what specific needs are unmet by mainstream platforms. Are these related to negative online experiences, safety concerns, or cyberbullying? Many of these issues are not unique to PwDs, so clarifying what makes this population’s needs distinct would strengthen your argument.

Your point about co-design is convincing and draws on the important work of previous scholars such as Elizabeth Ellcessor’s work (Restricted Access), you may want to consider reading this which examines how form, experience, and the role of designers shape digital spaces for PwDs and when one aspect is missing how it creates unique barriers for PwDs.

Response 2: Thank you. Based on reviewers feedback, we introduced ConnectUp earlier and hopefully more clearly; we explained what makes ConnectUp distinct from other platforms; clarified why PwD and carers need a separate digital space for social connection and physical activity (PA); supported the argument with existing research on how PwD and carers use (or struggle with) mainstream platforms and referred to Elizabeth Ellcessor’s work (Restricted Access). We highlighted our changes in the introduction and discussion and we cite them below:

“Digital tools have increasingly been used to promote PA and build social networks, yet existing mainstream platforms (e.g., Facebook, Meetup, Strava) often fall short for PwD and carers. Research suggests that while these platforms may support connection, they frequently fail to provide safe, accessible, and inclusive environments tailored to the distinct needs of users with disabilities and carers (Alhaboby et al., 2019; Ellcessor, 2016).”

“To address these barriers, we developed a prototype of ConnectUp, a co-designed digital platform built specifically for adult PwD and carers. Unlike existing apps that target broad populations, ConnectUpwas designed to reflect the lived experiences, communication styles, and accessibility requirements of its users.

Co-design was central to ConnectUp’s development, enabling meaningful participation from PwD and carers throughout the design process.

“A key design decision in the development of the platform was to limit access to persons with disabilities (PwD) and carers. This choice emerged directly from user consultations, where participants expressed a strong preference for a dedicated space tailored to their specific needs and lived experience. While concerns have been raised about potential exclusion, participants emphasized that existing mainstream platforms often fail to provide environments where they feel fully understood or adequately supported. Aligned with other researchers, we acknowledge the complexity of dedicated versus mainstream platforms and the potential for unintended consequences of separate spaces (Kaur & Saukko, 2022). However, the creation of a closed, community-specific platform was intended to promote psychological safety, foster peer-to-peer support, and facilitate communication without the burden of explaining disability-related experiences or carer-experiences. Future iterations may consider broader integration; however, the initial focus remains on addressing the unique priorities identified by the core user group.”

Comment 3: Methods
In Section 2.3, you note that workshops 5–6 used a draft ConnectUp prototype, but provide little detail about what the prototype entailed. What did participants do with it? What features were tested? How long did they interact with it? Did they receive guidance or explore it freely? This goes back to the original point of providing context to ConnectUp for the following arguments to make sense to the reader.

Response 3: Thank you for this valuable comment. We have now added clarification to the manuscript to explain that participants did not interact directly with the app. Rather, they were shown a prototype on screen during an online meeting. The following text has been added to provide this detail: “The prototype draft was showed as clickable mobile phone application generated in Figma. Workshop facilitator showed user’s journey clicking screen-by-screen asking workshop participants to share their feedback and reflections on the demonstrated prototype including look-and-feel, wording, and functionality.”

Comment 4: Much of the methods section focuses on which author facilitated which activity (writing in this way also makes the narrative read in a clunky way), but this does not help readers understand the process from the participants’ perspective. More detail is needed on how participants engaged in the workshops and what their involvement was like.

Response 4: Very good point, we removed most of the initials in the brackets and we no longer repeat them so often. We added: “Participants were asked about their experiences, beliefs, and attitudes towards improving social connection, PA, and general well-being. They were also asked to elaborate on online platforms that they used and to share their preferences and expectations for online tool supporting these needs. The workshops were very informal, and participants were encouraged to share their experiences and to reflect on how online platform could support their needs to improve social connection, inclusion, and encourage them to engage in shared PA with other PwD and carers.” To explain how participants engaged in the workshops and what their involvement was like.

Comment 5: You mention that Carers WA was the recruitment site but do not explain why this setting was chosen. Given that a member of the research team (PR) is also a manager at Carers WA, how did this influence recruitment and participants’ willingness to participate?

Response 5: We added a sentence that explains that: “Carers WA was the lead organization responsible for delivering this project and developing the ConnectUp platform.”

Comment 6: The description of the research team’s identities and skills (e.g., DK as a behavioural scientist, CT as an occupational therapist) is valuable but currently disconnected from a discussion of reflexivity. Clarify how the team’s backgrounds shaped the study design, participant interaction, rapport, recruitment, and potential biases during facilitation and your interpretation of the findings. This is important as only one member engaged in member check.

Response 6: Thank you for this thoughtful observation. We agree that the inclusion of the research team's identities and skills is important, and we appreciate the opportunity to clarify how these shaped the study through a reflexive lens. We have now revised the manuscript to more explicitly connect the research team’s backgrounds to their roles in the study, acknowledging how this influenced design decisions, participant rapport, data interpretation, and potential biases. We added: “The involvement of researchers with lived experience of disability (SJe, AS) and staff from Carers WA (SJe, JH, PR) informed the design of the co-design workshops to ensure cultural and contextual relevance. This also supported more meaningful participant engagement and trust-building during recruitment and facilitation” and “The mix of disciplinary expertise, occupational therapy, behavioral science, health psychology, and physiology, shaped the development of the interview guide, coding framework, and interpretation of the data. For example, CT brought clinical insight into how disability and caregiving influence daily life, while EM and DK contributed knowledge of behavior change strategies. This diversity enriched the analysis but also required ongoing reflection during team discussions to address disciplinary biases and avoid privileging one perspective.”

Comment 7: There is also no mention of research ethics and a note on this would help the reader to understand how you ensured ethical research took place and where ethical approval was obtained.

Response 7: Thank you, we added: “The study was approved by the Institutional Review Board by the Human Research Ethics Committee at Curtin University (HRE2021-0678), Perth, Australia. All participants provided informed consent prior to participation.” At the end of the Participants section. Previous version included info about the ethics approval at the end of the manuscript. Now, we list it in both places.

Comment 8:

Results
Although the study’s focus is on co-designing
 ConnectUp, the results only briefly address this. Most findings describe broader themes such as online safety and inclusion. It may be worth reconsidering whether the paper should be framed more broadly around the features and social affordances that PwDs and carers want from digital platforms to support PA and social connectedness. You might find relevant perspectives in Kaur & Saukko’s work on how PwDs use mainstream platforms instead of those explicitly designed for them, sometimes dedicated spaces can unintentionally reinforce exclusion as many PwDs are found helping each other overcome digital design barriers in support groups for people with disabilities.

Response 8: Thank you for this helpful suggestion. In response to this comment—as well as a related point raised by the other reviewer, we have added a section to the Discussion that elaborates on the rationale for designing a platform specifically tailored to the needs of PwD and carers. We agree that the themes identified, such as online safety, inclusion, and accessibility, extend beyond the technical features of the ConnectUp prototype and reflect broader social affordances valued by PwD and carers in digital platforms. While our primary aim was to inform the co-design of ConnectUp, we acknowledge that these findings also speak to more general expectations of inclusive digital environments. To address this point, and ensure alignment between the focus and findings, we have added clarification in the Discussion to reflect how the results contribute both to the design of ConnectUp and to wider conversations about digital inclusion and social connection. We have retained the current framing to maintain consistency with the study’s applied, co-design focus, but we appreciate the broader implications and have noted this in our revised discussion. Thank you also for the suggestion to engage with Kaur & Saukko’s work—we have reviewed it and now reference it to acknowledge the complexity of dedicated vs. mainstream platforms and the potential unintended consequences of separate spaces. We added: “A key design decision in the development of the platform was to limit access to persons with disabilities (PwD) and carers. This choice emerged directly from user consultations, where participants expressed a strong preference for a dedicated space tailored to their specific needs and lived experience. While concerns have been raised about potential exclusion, participants emphasized that existing mainstream platforms often fail to provide environments where they feel fully understood or adequately supported. Aligned with other researchers, we acknowledge the complexity of dedicated versus mainstream platforms and the potential for unintended consequences of separate spaces (Kaur & Saukko, 2022). However, the creation of a closed, community-specific platform was intended to promote psychological safety, foster peer-to-peer support, and facilitate communication without the burden of explaining disability-related experiences or carer-experiences. Future iterations may consider broader integration; however, the initial focus remains on addressing the unique priorities identified by the core user group.”

Comment 9: There is also occasional confusion due to the combined analysis of two population groups: PwDs and carers, without clear distinctions between them. At times when mentioning ‘participants’ it is unclear which group is being referenced. Moreover, the diversity within the PwD community and within one community (intra-disability diversity) is not acknowledged; P. Tsatsou’s work may be helpful here. Treating PwDs as a single, homogenous group while only acknowledging the conditions risks oversimplification and shows a lack of awareness of literature in disability studies.

Response 9: Thank you for this insight. It appears there may be some misunderstanding regarding the composition of our participant group. As clarified in the Participants section, these are not two distinct groups, but rather a single group comprising both PwD and carers, some of whom have overlapping identities (meaning are PwD and carer): “Six workshops were conducted with 17 participants (male = 8, female =9; PwD = 6, carer = 6, and PwD and carer = 5) aged 20 to 73 years (M = 52.94, SD = 15.21). Participants' disabilities included physical, intellectual, and mental health conditions. Participants who were carers had different caring responsibilities ranging from ad-hoc caring to full time caring roles.” We know that carers are statistically more likely to have a disability than the general population. Studies have shown that a significantly higher percentage of carers report having a disability, poor health, or limitations in their daily activities compared to non-carers.  We agree that there is considerable diversity within the PwD community, and we are aware of the risks associated with treating it as a homogenous group. We also appreciate the reference to Tsatsou’s work, which offers valuable insight into intra-disability diversity. In this study, our aim was to identify shared priorities and challenges across a diverse group of participants in the context of co-designing a digital platform. While we recognise that lived experiences differ significantly between individuals and disability types, our focus was on collective themes relevant to the development of an inclusive tool. As such, we have intentionally presented findings at a cross-disability level, consistent with the applied nature of the project. We have included a brief note in the limitations to acknowledge the broader issue of intra-group diversity and suggest this as an area for future research. “This study did not explore intra-disability diversity in depth, and future research is needed to examine how differing lived experiences may shape preferences and needs.”

Comment10: It would be helpful to clarify whether individual workshops included both PwDs and their carers together. Some quotes (section 3.1.3) suggest that carers were often parents who are speaking about young children with disabilities, but there is little critical analysis of how carers’ perceptions may frame disability in infantilising or limiting ways, despite PwDs’ own sense of agency and capability. This tension is well documented in disability studies and should be critically addressed.

Response 10: Thank you for this thoughtful comment. We have reviewed the section in question and would like to clarify that the workshops included both PwD and carers together, as described in the Participants section. None of the participants were parents of young children with disabilities. Only one carer participant had an adult daughter with disability, and her reflections were framed accordingly.

We acknowledge the important tensions raised in disability studies literature regarding how carers’ perspectives can sometimes conflict with the agency of PwD. However, given the composition of our sample and the applied focus of this co-design study, a critical exploration of these dynamic falls outside the scope of the current paper. We have added a brief clarification in the Methods section to avoid any confusion about participant backgrounds. We added: “Workshops included a mix of carers, PwD and people identifying as both.”

Comment 11: Some themes, such as “fear of miscommunication” (section 3.1.2), appear unrelated to the unique needs of PwDs and carers and could apply to digital platform users more broadly. Similarly, section 3.2 notes that the platform “could be inclusive and guided by the person’s interests and experiences,” but without clear context about the platform’s features, this is vague and makes little sense to the reader.

Response 11: Thank you for this observation. We agree that concerns such as fear of miscommunication may not be unique to PwD and carers and could indeed apply to digital platform users more broadly. However, we chose to retain this theme as it emerged consistently across our data and holds relevance in the context of designing inclusive digital spaces, especially given that effective communication is often cited as a key barrier to participation for both PwD and carers. Acknowledging these broader themes ensures the platform design addresses not only disability-specific needs but also more universal usability concerns that, if unaddressed, may disproportionately affect marginalised users. We agree that the sentence in Section 3.2 lacked sufficient context and may have appeared vague. We have revised the text to: “The participants suggested that the platform should be inclusive by offering personalized content tailored to users' interests, goals, and lived experiences, for example, by recommending relevant activities or groups”

Comment 12: Discussion
It is not clear what the article’s original contribution is or what new insight it offers that advances debates about accessible digital platform design for promoting PA among PwDs and carers. Please clarify the main argument and how it builds on, contrasts with, or extends existing scholarly literature. I would also suggest updating the abstract in line with this.

Response 12: Thank you for this feedback. We appreciate the opportunity to clarify the article’s contribution. In response to this comment we added a section at the start of the discussion that highlights article’s original contribution and what new insight it offers that advances debates about accessible digital platform design for promoting PA among PwDs and carers: “This study adds to the growing body of literature on accessible digital design by centering the perspectives of both PwD and carers in the co-design of a platform aimed at promoting PA and social connection. While prior research has explored barriers to PA participation and use of digital platforms separately, few studies have addressed the intersection of these issues through a co-design approach that includes both groups. Our findings offer novel insights into user preferences, trust-building features, and perceived affordances that are often overlooked in mainstream platform development”

Comment 13: You note that the results did not reflect a diversity of views, but do not explore why. I have the impression this was due to the sampling strategy, or was this due to other factors which are not discussed? Discussing this would help situate the findings and acknowledge potential limitations with clear explanations in a convincing way.

Response 13: Thank you for this comment. To clarify, we did not state that the results lacked a diversity of views, but rather that the sample was not as diverse as anticipated—specifically noting that certain groups (e.g., individuals with limited hearing or vision) were not represented. This was likely due to a combination of factors, including our recruitment strategy. We have now added a sentence to the Limitations section to acknowledge this. Revised sentence reads: “Although the aim was to recruit a diverse sample of PwD and carers in terms of disability type, the included sample was not as diverse as anticipated (e.g., people with limited hearing or limited vision did not participate) likely due to the recruitment approach which may have influenced the range of perspectives captured.”

Comment 14: Conclusion
In the conclusion, you write about the need for “genuine inclusion … essential for fostering motivation and enabling authentic social connection,” but it is unclear who these connections are between: PwDs and other PwDs, carers and PwDs, or wider community members? Prior work (see, Kaur & Saukko) suggests that even when digital platforms are technically accessible, social connections are still hindered by broader societal attitudes towards disability when the connection is between PwDs as well as between PwDs and able-bodied people. Will the ConnectUp platform address such existing barriers presented in current research?

Response 14: Thank you for this important observation. We agree that fostering meaningful social connection is a complex process, particularly in the context of broader societal attitudes toward disability. In this study, participants spoke about the value of connecting with others who share similar lived experiences, including both other PwD and carers, as a means of feeling understood and supported. While the ConnectUp platform was primarily designed to facilitate connection among PwD and carers, it is envisioned as an inclusive space that could eventually support broader community engagement. We acknowledge that technical accessibility alone does not resolve social barriers, as highlighted in Kaur & Saukko’s work. While ConnectUp may not fully overcome these systemic challenges, it was developed with the aim of creating a safer, more supportive environment where users feel comfortable initiating and sustaining connections. We have clarified this point in the Conclusion and now acknowledge that broader attitudinal barriers remain a key consideration for future development and evaluation of such platforms, adding: “We recognize, however, that broader societal attitudes and systemic barriers remain challenges that technical accessibility alone cannot fully resolve.”

Comment 15: Finally, the paper implies that PA is an activity to be sought externally, yet many PwDs and carers may integrate PA into daily routines at home, at work, or in social contexts. How did participants conceptualise PA and what does this look like for PwDs and carers? Although the interview guide included relevant questions about this, I anticipate the workshops probably answered this, but the article does not clearly present these insights. Therefore overall the connection between PwDs, carers and PA is loose.

Response 15: Thank you for this valuable observation. We appreciate the importance of clarifying how participants conceptualized physical activity (PA) within the study. To address this, we would like to highlight that Section 3.3 of the manuscript is dedicated to exploring participants’ diverse perspectives on PA, including the recognition that PA is often integrated into daily routines, not just sought externally. This section details the varied experiences of PwD and carers with PA, encompassing positive and negative associations, environmental, social, and financial barriers, as well as motivational factors. We believe this adequately addresses the nuanced ways in which PA is understood and experienced by the participants.

Comment 16: Overall, the study’s findings about what PwDs and carers want in a digital platform appear transferable to mainstream platforms and apps. A revised conclusion could situate these insights within broader discussions about accessibility, digital design, social relations, and everyday wellbeing, which could help clarify the article’s contribution and relevance.

Response 16: Thank you for this insightful comment. We agree that many of the study’s findings resonate with broader themes in accessibility and digital design. To clarify the unique contribution of our work, we revised the Conclusion section. We added: “These priorities speak to wider concerns in digital design, where accessibility and inclusion are increasingly recognized not only as technical requirements but as fundamental to promoting digital equity and social participation.” And “ More broadly, these findings highlight how digital platforms can play a role in shaping wellbeing, not only by facilitating access to PA but also by enabling meaningful social interaction. By aligning design features with the lived experiences of PwD and carers, mainstream platforms and apps can move toward more inclusive digital environments that foster both autonomy and belonging.”

Comment 17: Overall, I appreciate it is not easy to bring many elements together in a single article, and it is therefore down to the authors to rethink how to reframe the main argument so the contribution is clearer. Whether it is making ConnectUp the focus or broader issues around digital accessibility and co-design. I wish the team the best with figuring this out and hope the comments here support you with the revision.

Response 17: Thank you very much for your thoughtful feedback and encouragement. We appreciate your recognition of the complexities involved in integrating multiple elements into a single article. Your suggestions have been extremely helpful in guiding us to clarify and strengthen our main argument. We truly value your support in improving our work and hope that the revised version is more informative, better structured, and reads more clearly overall.

Reviewer 2 Report

Comments and Suggestions for Authors

I would like to see a section on why the social platform was limited to PwD and their carers. Why wasn't it opened to people without disabilities? This restriction may actually reinforce the loneliness of PwD and their carers as they are 'only good enough' to socialize within their own community.

Author Response

Comment 1: I would like to see a section on why the social platform was limited to PwD and their carers. Why wasn't it opened to people without disabilities? This restriction may actually reinforce the loneliness of PwD and their carers as they are 'only good enough' to socialize within their own community.

Response 1: Thank you for raising this important point. We understand the concern that creating a platform exclusively for persons with disabilities (PwD) and their carers might unintentionally contribute to a sense of segregation. During the design phase, we consulted members of the PwD community and their carers. The overwhelming feedback was a desire for a dedicated space where their unique experiences, needs, and ways of communicating could be fully understood and respected without the need for explanation or justification. Many felt that mainstream platforms often lack this level of understanding, and that creating a space tailored to their realities could foster more meaningful connections and reduce feelings of isolation. We altered the introduction to explain the rationale and we also added the section below to our Discussion section:

“A key design decision in the development of the platform was to limit access to persons with disabilities (PwD) and carers. This choice emerged directly from user consultations, where participants expressed a strong preference for a dedicated space tailored to their specific needs and lived experience. While concerns have been raised about potential exclusion, participants emphasized that existing mainstream platforms often fail to provide environments where they feel fully understood or adequately supported. Aligned with other researchers, we acknowledge the complexity of dedicated versus mainstream platforms and the potential for unintended consequences of separate spaces (Kaur & Saukko, 2022). However, the creation of a closed, community-specific platform was intended to promote psychological safety, foster peer-to-peer support, and facilitate communication without the burden of explaining disability-related experiences or carer-experiences. Future iterations may consider broader integration; however, the initial focus remains on addressing the unique priorities identified by the core user group.”

Round 2

Reviewer 1 Report

Comments and Suggestions for Authors

With the addition of new references in the list at the end of the document, there are now two references appearing in the same line and just needs a minor edit and proof read. 

Thank you for the opportunity to review this article which has addressed my previous comments, I believe this has enhanced the final quality of the paper. 

Reviewer 2 Report

Comments and Suggestions for Authors

Thank you for making the revisions requested.